Transfer learning-based English translation text classification in a multimedia network environment

Zheng Danyang danyang.zheng@tju.edu.cn
School of Foreign Languages, Tianjin University , Tianjin , China
Mumtaz Shahid
Electronic publication date: 2024 Jan 31
Publication date: 2024
Volume: 10
Electronic Location ID: e1842
Received 2023 Aug 29; Accepted 2024 Jan 9
Copyright: © 2024 Zheng
Copyright year: 2024
Copyright holder: Zheng
License: This is an open access article distributed under the terms of the Creative Commons Attribution License, which permits unrestricted use, distribution, reproduction and adaptation in any medium and for any purpose provided that it is properly attributed. For attribution, the original author(s), title, publication source (PeerJ Computer Science) and either DOI or URL of the article must be cited.
License URL: https://creativecommons.org/licenses/by/4.0/

Keywords: Multimedia network technology, Transfer learning, English translation, Text classification

Funding: Tianjin Philosophy and Social Science TJZWQN18-003 This work is supported by the Tianjin Philosophy and Social Science Project (TJZWQN18-003). The funders had no role in study design, data collection and analysis, decision to publish, or preparation of the manuscript.

==============================
In recent years, with the rapid development of the Internet and multimedia technology, English translation text classification has played an important role in various industries. However, English translation remains a complex and difficult problem. Seeking an efficient and accurate English translation method has become an urgent problem to be solved. The study first elucidated the possibility of the development of transfer learning technology in multimedia environments, which was recognized. Then, previous research on this issue, as well as the Bidirectional Encoder Representations from Transformers (BERT) model, the attention mechanism and bidirectional long short-term memory (Att-BILSTM) model, and the transfer learning based cross domain model (TLCM) and their theoretical foundations, were comprehensively explained. Through the application of transfer learning in multimedia network technology, we deconstructed and integrated these methods. A new text classification technology fusion model, the BATCL transfer learning model, has been established. We analyzed its requirements and label classification methods, proposed a data preprocessing method, and completed experiments to analyze different influencing factors. The research results indicate that the classification system obtained from the study has a similar trend to the BERT model at the macro level, and the classification method proposed in this study can surpass the BERT model by up to 28%. The classification accuracy of the Att-BILSTM model improves over time, but it does not exceed the classification accuracy of the method proposed in this study. This study not only helps to improve the accuracy of English translation, but also enhances the efficiency of machine learning algorithms, providing a new approach for solving English translation problems.

Introduction

The development of multimedia network technology has led people to increasingly encounter the need for cross lingual communication and exchange, especially the widespread application of English translation technology in modern society (Sharma & Singh, 2023). However, facing the constantly growing massive amount of data, diverse linguistic text information, and multimodal multimedia data, how to quickly and accurately classify texts has become an urgent issue in translation text processing. In response to this issue, transfer learning has been proposed as a new learning method, which learns knowledge from the source domain and transfers it to the target domain to improve the learning effectiveness of the target task (Ferreira-Mello et al., 2019). The research on translation text classification theory and methods based on transfer learning has become another highly anticipated direction in this field (Hung & Chang, 2021). Although deep learning methods have achieved significant results in the field of text classification in recent years, they still have certain limitations for massive data and text classification in multiple languages (Lu et al., 2015).

This study will continue the research approach based on transfer learning, further inheriting the theories of incremental learning and adaptive learning, and proposing a translation text classification method based on deep neural network structure (Kraus & Feuerriegel, 2017). Among them, we learn common features in the source domain in advance and transfer them to the target domain through transfer learning to achieve efficient learning of the target domain text classification task. At the same time, we propose incremental learning and adaptive learning strategies to address the bottleneck problem of massive data (Ghorbanali, Sohrabi & Yaghmaee, 2022), in order to ensure the efficiency and accuracy of model learning while processing text classification tasks in real-time and adaptively. The fusion of incremental learning and adaptive learning strategies in this article can help models quickly learn and adapt to new data, improving the speed and accuracy of text classification task processing (Gupta & Jalal, 2022). In the future, we will continue to explore more efficient text classification methods and theories, expand their applications in multiple fields and languages, and provide strong support for building more intelligent and efficient translation text classification systems (Cao, 2020).

As a learning method, transfer learning can apply learned knowledge and experience to new tasks (Nurnoby, El-Alfy & Luqman, 2020). In English translation text classification, transfer learning can improve the classification effect by using the knowledge and experience gained in other related fields. Therefore, an English translation text classification method based on transfer learning is an effective way to solve this problem. The research framework is shown in Fig. 1.

Figure 1 Research framework diagram.

This research is a further extension of transfer learning technology based on multimedia internet technology combined with English translation text classification and a new text classification method. The main contributions of this research are as follows:

1) A new method for text classification is created by integrating modern multimedia network technology with transfer learning.

2) The text classification methods used in the Bidirectional Encoder Representations from Transformers (BERT) model, the attention mechanism and bidirectional long short-term memory (Att-BILSTM) model, and the transfer learning based cross domain (TLCM) model are deconstructed and merged, and a new technological convergence framework for English translation text classification is proposed.

3) The advantages of the proposed method in this study are verified, and a usage interval with high classification accuracy is found, providing a foundation for its practical operation.

The structure of this article is as follows. The first part introduces the background, purpose and framework of this study. In the second part, the relevant previous research is elaborated, analyzed, and summarized. The third part explains the transfer learning technology, the BERT model, the Att-BILSTM model and the TLCM model in multimedia network technology and their theoretical basis. In the fourth part, the above models are combined with a transfer learning method based on multimedia network technology, an English translation text classification framework is established, and then the requirements to complete the label classification are analyzed. In the fifth part, experiments are designed to verify the relationship between the methods established in this study and the BERT model, Att-BILSTM model, and TLCM model, and impact analysis is completed to obtain complete research results.

Literature review

In recent years, some scholars at home and abroad have conducted relevant research on the topic of “English translation text classification based on transfer learning in the context of media network technology” and made some important conclusions. Rojas-Barahona (2016) trained a CNN based neural model on a large dataset of hate tweets in selected major languages, and then retrained it on a small transliteration dataset in the same language to obtain the most representative combination of input settings, which helps to achieve state-of-the-art hate speech detection by swapping code on Twitter for multilingual short texts. Giorgi & Bader (2018) analyzed to what extent transfer learning improved BNER’s latest results, and found that transferring deep neural networks (DNNs) trained on large, noisy SSCs to smaller but more reliable GSCs significantly improved BNER’s state-of-the-art results. Compared to the latest baseline evaluated on 23 GSCs covering four different entity classes, the average error of transfer learning decreased by approximately 11%. Shin et al. (2016) utilized three important but previously insufficiently researched factors in applying deep convolutional neural networks to computer-aided detection problems to explore and evaluate different CNN architectures. The model under study contains 500 to 160 million parameters and has different levels. López-Sánchez, Arrieta & Corchado (2019) proposed a text classification method based on word2vec and transfer learning in their research on English translation text classification in a media network environment. They found that a pretrained word2vec model can effectively capture the semantic information of words, and transfer learning can improve the classification accuracy. Winata (2021) proposed a multimedia text classification method based on a deep neural network and transfer learning. Their research results showed that through a pretrained text representation model and transfer learning, efficient and accurate English translation text classification can be achieved in a media network environment. Cicero et al. (2017) studied the application of a multimedia text classification method based on transfer learning in a social media environment. They found that transfer learning can effectively solve the problem of domain adaptation in social media data, thus improving the performance of English translation text classification.

Fundamentals of research theory

Transfer learning based on multimedia network technology

In the multimedia network technology environment, transfer learning plays an important role in English translation text classification (Luo, Wen & Tao, 2017). Transfer learning is a machine learning method that improves performance on target tasks by transferring learned knowledge and experience to different tasks or domains (Kowsari, Heidarysafa & Barnes, 2019). In traditional English translation text classification, a large amount of annotated data and complex feature engineering are often required to train the model. However, this faces some challenges in the multimedia network technology environment. Multimedia network technology can generate a large amount of unstructured data, such as images, audio, and video. This type of data may be difficult to handle for traditional text classification methods (Wang, Xie & Chen, 2019).

In English translation text classification tasks, we can use these pre-trained models as feature extractors to convert multimedia data into feature vectors. Then, we can use transfer learning methods to apply these features to the target task for classification. Transfer learning can transfer existing knowledge to new tasks by sharing some or all model parameters, thereby reducing the need for annotated data and improving classification performance. In multimedia network technology, transfer learning can be used to solve the following problems:

1) Insufficient datasets: For certain multimedia network technology tasks, it may be difficult for the model to fully learn and generalize due to sample or dataset scarcity. By using transfer learning, data from other fields or tasks can be used to help model learning and extract useful features and knowledge.

2) Model initialization and parameter optimization: In multimedia network technology, using pretrained models or migrating existing models can help quickly initialize and optimize model parameters. Through transfer learning, the weights and characteristics of a pretrained model for a related tasks can be used to initialize the target task model to speed up training and improve the performance.

3) Feature extraction and representation learning: Feature extraction and representation learning of multimedia data are important tasks in multimedia network technology. Through transfer learning, models trained on large multimedia datasets can be used as feature extractors or feature generators to obtain better multimedia data representations and improve the performance of the models in related tasks.

4) Cross-domain adaptation: In multimedia network technology, there may be differences in different network environments, devices, and application scenarios, resulting in models trained in one domain being unable to be directly applied to other domains. Through transfer learning, knowledge and experience in one domain can be migrated to the target domain through domain adaptation, allowing adaptation to different network environments and application scenarios.

BERT basic transfer learning model

Bidirectional Encoder Representations from Transformers (BERT) is a pre-trained language model based on the Transformer architecture, which has achieved great success in natural language processing tasks. Transfer learning is a method of utilizing knowledge already learned on one task to accelerate the learning of another task. The English translation text classification method based on transfer learning can utilize the BERT model to improve the performance of classification tasks (Marulli, Verde & Campanile, 2021). In transfer learning, we use the pre-training results of the BERT model on large-scale corpora as initialization parameters and fine tune the data for specific tasks. For the English translation text classification task, fine tune the BERT model according to the task requirements to obtain a model that is more suitable for the task. The fine-tuning process includes two stages: pre-training and fine-tuning. In the pre-training stage, we use the BERT model for unsupervised pre-training on large-scale corpora, with the aim of enabling the model to learn better language representations. In the fine-tuning stage, we use task specific annotated data to perform supervised fine-tuning of the BERT model on task specific objective functions, in order to better adapt the model to specific tasks.

The BERT model is a large-scale pretrained language model based on transformer encoders proposed by Google in 2018. Unlike traditional language models (Feng & Chaspari, 2020), BERT uses more pretrained data and a deeper network model structure. BERT also models more complex language features to generate vector representations of deep upper and lower bounds, effectively solving the problem of polysemy. The encoder operation process is shown in Fig. 2.

Figure 2 Schematic diagram of BERT encoder operation process.

In terms of the impact of distribution bias between domains, specifically, for each sample x in the training set, the label z ∈ 0,1 indicates whether a sample belongs to the source domain or the target domain. A domain discriminator D is introduced to determine the feature representation h = G(x;θg) probability P of source domain P(0| x;θg). This is a two-category problem, and its corresponding loss function is defined as (Luna-Jiménez et al., 2021):

(1) LD(θd)=−∑i=1nxilog(P(0/G(xi;θg);θd))+log(P(1/G(xi;θg);θd))

where θg represents the feature encoder parameters and θd represents the domain discriminator parameter. On the one hand, to accurately determine whether each sample is from the source domain or the target domain, the domain discriminator should constantly minimize the loss function. On the other hand, to generate a domain-independent feature representation, the feature encoder should attempt to fool the discriminator to constantly maximize the loss function. The encoder submodule is shown in Fig. 3.

Figure 3 Schematic diagram of BERT encoder submodule.

Attention mechanisms have now become an important concept in the field of neural networks. In the field of artificial intelligence (AI), attention has become an important part of the neural network structure and has a large number of applications in natural language processing, image recognition, speech recognition and other fields. An attention model aims to alleviate these challenges by allowing the decoder to reference the entire encoded input sequence. The main idea is to introduce attention weights to the input sequence. During decoding, closely related information in the input sequence is selected based on the attention weights, while useless information is ignored, thereby reducing the computational burden of the neural network. Specifically, given the input sequence representing H = [h1,h2,…,hN] and a task-related query vector q, the probability of selecting the i-th input information is αi (Zhang, Lertvittayakumjorn & Guo, 2019).

(2) α=softmax(s(hi,q))=exp(s(hi,q))∑j=1Nexp(s(hj,q))

where αi is the attention distribution and s(xi, q) is the attention scoring function. The following method can be used:

(3) s(hi,q)=vTtanh(Wxi+Uq)

where W, U, and v are learnable network parameters. The attention distribution can be interpreted as the degree of attention given to each piece of information given the query vector q related to the task. Then, we use a soft attention mechanism to summarize the representation of the input sequence:

(4) v=∑iNαihi

Classification method based on Att-BILSTM

The transfer learning based English translation text classification method is an important research direction in the field of text classification in the multimedia network technology environment. To better address this issue, the article proposes a classification method based on the attention mechanism and bidirectional long short term memory (Att-BILSTM) network. The core idea of this method is to combine attention mechanism and bidirectional long short-term memory network. Firstly, this method utilizes the idea of transfer learning to pre-train an English translation text classification model from one or more relevant domain datasets. Then, by introducing attention mechanisms, the model can automatically learn and focus on important information in the text, improving the accuracy and robustness of classification results.

The Att-BILSTM model consists of two key components: attention mechanism and bidirectional long short-term memory network. The attention mechanism allows the model to dynamically allocate different attention weights when processing each word in a text sequence. In this way, the model can focus more on content related to classification tasks and ignore irrelevant information. The bidirectional long short-term memory network can effectively capture contextual information in text sequences, helping the model better understand semantic and structural features. Let H be an output vector, which is a matrix composed of H = [h1, h2,…, hT] that is generated in the LSTM layer, where T is the total number of words in the sequence. The calculation of the feature vector for sentence representation is as follows (Acheampong, Wenyu & Nunoo-Mensah, 2020):

(5) M=tanh(H) β=softmax(wTM) r=HαT

where H ∈ Rdw*T, W, r, and β are d-, T-, and d-dimensional vectors, respectively. The sequence features are represented as:

(6) h∗=hanh(r)

Attention mechanisms can help models automatically focus their attention on key information. In Att-BILSTM, the attention mechanism can assign a weight to each input position, allowing the model to better focus on key parts and use softmax classifiers to predict labels from the discrete set of category Y in sentence S. The classifier takes hidden state h* as input (Duarte & Berton, 2023):

(7) P^(y∣S)=softmax(W(S)h∗+b(S)

(8) y^=arg​​maxyp^(y/S)

where t ∈ Rm is the unique heat representation, y ∈ Rm is the estimated probability multiplied by the softmax score for each category (m is the number of target categories), and λ regularizes the hyperparameters for L2. According to the needs of the task, different operations can be used in the output layer. For example, for sequence annotation tasks, a label classification layer (softmax classifier) can be used to predict the labels at each position. The Att-BILSTM model can fully utilize contextual information and focus on task-related parts by integrating a bidirectional LSTM and an attention mechanism. This can improve the performance of the model in multiple natural language processing tasks. The depth domain adaptation model structure is shown in Fig. 4.

Figure 4 Schematic diagram of deep domain adaptation model structure.

TLCM improved pretransfer learning model

Transfer learning is widely studied and applied in the field of computer science, especially for natural language processing tasks, which is of great significance. In traditional transfer learning methods, large-scale annotated texts from the source and target domains are often used to train transfer learning models. However, in the topic I am studying, there is often a significant difference between the source domain and the target domain, for example, the source domain may be news reporting while the target domain may be social media comments. This difference leads to a decrease in the performance of traditional transfer learning models. The core idea of the transfer learning cross domain (TLCM) model is to introduce intermediate domain data for transfer learning. Intermediate domain data refers to data that has a certain connection between the source domain and the target domain, for example, it can be blog articles in online communities. The text classification framework is shown in Fig. 5.

Figure 5 TLCM text classification framework.

Based on the transformer structure and Bi-LSTM-CNN text class model, the semantic information of the current character context is fully considered to learn the global structural information of the text transformer Bi-LSTM-CNN (TLCM) model framework.

Traditional word vector representation methods such as word2vec or GloVe require word segmentation technology to input the input document according to the word level after data preprocessing. The embedding structure of the ALBERT model trains on the input document at the sentence level by inputting the original text vector into the transformer structure at the word level, achieving dynamic word vector representation of medical text. In the ALBERT pretrained language model for transfer learning, the text input part is the Chinese health question text W with a length of n, which is composed of a series of characters in each health question document, as shown in the following formula (Li et al., 2019):

(9) W={W1,W2,⋯,Wn}

To effectively represent Chinese health question text as a text vector, the ALBERT pretrained language model for transfer learning is built, and the 巸 padding layer of the network structure is fine tuned for initialization. For the input health question text description W, each word in the text ν can be mapped into a word vector initialization representation ei by embedding the embedding structure of the original ALBERT model (Xiong et al., 2023).

(10) ei=E(wi)

In the equation E∈R∣V∣×k, |v| is the size of the vocabulary, and k is the dimension of the word vector.

To meet the multilabel classification task demand of health questions describing an indefinite number of labels in the text, the label prediction network module designs a multilayer perceptron with two fully connected layers to build a multilabel classifier. The sigmoid activation function is used to convert the output value of the network layer into the output probability to estimate the prediction probability of each label in the document. Its calculation formula is shown in the following formula (Çalışkan, 2020).

(11) y^=sigmoid(W2f(W1VT))

The numbers W1∈Rb×2k,W2∈Rb represent the parameter transformation matrices of the fully connected layer and the output layer, respectively. The model activation function f is ReLU, and the cross-entropy loss is used to realize the backpropagation training of the model. The update process is as follows (Weber et al., 2021):

(12) ζ=−∑i=1N∑j=1L⁡(yijlog(y^ij))+(l−yij)log(l−y^ij)

where N represents the number of training documents, L represents the number of labels corresponding to the documents, yij^∈[0,1] represents the prediction probability of each label corresponding to each text document, l indicates whether the predicted result of the i text document belongs to the j label, where 1 indicates “yes”, and 0 indicates “no”.

By introducing intermediate domain data, the TLCM model can better utilize the connection between the source and target domains, and improve the performance of transfer learning models in the target domain. Meanwhile, the TLCM model can also adapt to different multimedia network technology environments, such as adapting to the characteristics of social media comments.

Establishment of the batcl transfer learning model

Data preprocessing

Text classification requires the process of automatically classifying unstructured text information according to given classification rules or benchmarks. Below is the definition of text classification for this study; n represents the number of documents, D represents the set of documents, i.e., D = {d1, d2, d3…, dn}, m represents the number of document categories, and Z represents the set of document data types, i.e., Z = {z1, z2, z3…, zn}. The process of text classification can be described by the following formula:

(13) Z=f(D,0)+ϵ

where f is the classification function and θ represents a set of parameters to be trained associated with the classification function f. The error ε refers to the difference between the classification result and the actual result, indicating that the classification function n is an approximation of the actual category.

Appear in the text to set the feature weight to 1. If it appears more than once, the feature weight is set to 1; otherwise, it is set to 0. Assuming that the feature vector corresponding to a certain text dk is (t1, t2, t3, …tm), the weight of BM is as follows:

(14) wki={1ti∈dk0otherwise

where wki represents the weight of the Boolean model, which is the weight of the feature term k in text d. Its biggest advantage is that the representation is simple and fast, while its disadvantage is that the representation is not sufficiently accurate. Thus, the importance of feature items to the text cannot be reflected, and the semantic information of the text cannot be expressed.

The most commonly used method for calculating weights is the text frequency–inverse document frequency (TF-IDF) method. TF-IDF is a statistical method whose main idea is that if a word feature appears frequently in a certain document but rarely in other documents, it is considered important for category differentiation. The formula for calculating the text frequency (TF) is as follows:

(15) TF(i,j)=ni,j∑knk,j

where i represents the variable of the feature word, j represents the variable of the document, and k represents the total number of documents. Therefore, nij represents the number of occurrences of the word feature in document dj, ∑knk,j and represents the sum of the total number of occurrences of all vocabulary in document dj.

Requirement analysis

Requirements analysis played a crucial role in my research. Through requirement analysis, I am able to identify the specific requirements and challenges in English translation text classification in the multimedia network technology environment. The study analyzed the demand for English translation text classification in the multimedia network technology environment. With the development of the Internet and multimedia technology, more and more English translated texts are widely disseminated and used on the internet. This requires automatic classification of these texts to facilitate information organization and retrieval. In addition, multimedia network technology has brought a large amount of multimodal text data, so in classification methods, it is necessary to consider how to effectively process and utilize this multimodal information.

We explored the requirements for English translation text classification based on transfer learning. Transfer learning refers to the technique of using knowledge learned from one field to help solve problems in another field. In English translation text classification, due to the similarity and shared knowledge between different fields, transfer learning can effectively utilize existing knowledge to improve classification performance. Therefore, the need for transfer learning is particularly important in this issue (Yadav et al., 2022). Corresponding problems and challenges were analyzed in response to these needs. Requirement analysis is one of the important stages of software development. In this article, the requirement analysis of English translation text classification based on the joint BERT model, Att-BILSTM model and TLCM model is completed based on multimedia network technology and the transfer learning method. The first step is to describe the problem. Our goal is to implement a system that can classify English text, targeting a segment or piece of English text.

This system needs to have functions for preprocessing, feature extraction, and classification of English texts. The preprocessing stage includes word segmentation, sentence segmentation, and stem extraction. The feature extraction stage uses BERT (a deep learning model) to encode English text into a fixed length vector representation. In the classification stage, the Att-BILSTM model is used, with the output of BERT as input. In addition, the system also needs to apply TLCM models trained on other tasks to English translation text classification tasks through transfer learning, in order to improve classification performance. In order to optimize the performance of the system, it is necessary to fine tune and optimize the hyperparameters of the TLCM model, and evaluate and optimize them using evaluation metrics such as accuracy, recall, and F1-score.

In addition to functional requirements, the system also needs to meet non functional requirements, including performance, availability, accuracy, scalability, and security. The system needs to be able to complete text classification tasks in a reasonable amount of time, while also having good scalability and efficiency for large-scale datasets. User usability is also very important, so the system needs to have good user friendliness and usability, while ensuring the accuracy of classification results. The system also needs to support flexible model extension and integration in order to achieve more functional extensions and improvements in the future. Finally, the system also needs to protect the security and privacy of user data, ensuring that the text data entered by users is not abused or leaked.

Label classification

The purpose of text label classification is to automatically classify a given text in order to better organize and manage a large amount of text data. By labeling text as specific categories or topics, it can improve the efficiency of text search, filtering, and browsing, and provide users with more accurate and personalized information. Using English translation to label and classify text is to better understand and organize text data, making it easier to search, manage, and analyze. By classifying text content, similar texts can be grouped together and assigned a label or keyword for each category, in order to quickly and accurately identify and retrieve the required information. This classification method can help users quickly obtain target information, while also promoting in-depth analysis and mining of text data. The basic classification method can use some common labels, as shown in Table 1. These labels can be adjusted and expanded according to different needs and fields to meet specific classification needs.

Table 1 Classification labels and their meanings.

Label	Meaning	Label	Meaning	Label	Meaning	
vn	Nominal verb	r	Pronoun	n	Common noun	
nt	Institution name	c	Conjunction	s	Locative noun	
an	Nominal word	nr	Name	nz	Other proprietary	
a	Adjectives	d	Adverb	w	Punctuation	
ad	Adverbial words	t	Time	f	Positional noun	
nw	Title of work	ns	Place name	s	Locative noun	
vd	Verb	u	Auxiliary words	v	Ordinary verb	
TIME	Time	q	Quantifier	xc	Other function words	
p	Preposition	PER	Name	m	Quantifier	
LOC	Place name	ORG	Institution name			

Using label classification can accelerate text data search, allowing users to quickly locate and access specific categories of text based on labels without having to browse the entire dataset. Label classification can help organize large amounts of text data, making it easier to manage. Categorizing text can better understand and grasp its structure and content, and improve the efficiency of information retrieval and text analysis. By summarizing and statistically analyzing texts with specific labels, relevant information about specific topics or fields can be obtained. At the same time, further analysis can be conducted, such as data mining and machine learning. Based on label classification results, personalized recommendation content can be provided to users. Based on user preferences and interests, recommend relevant text content to improve user satisfaction and experience. Through statistical analysis of label classification results, insights into text resources can be understood and reference basis can be provided for decision-making, supporting business development and optimization. Users can use tag keywords to search for relevant text, providing a more convenient and efficient way to obtain text. Label classification can also be used for organizing and managing translated texts, promoting the organization, sharing, and dissemination of knowledge.

Label classification is an effective way to quickly search for text data. By classifying text data according to different themes, content, or related attributes, search speed and accuracy can be improved. Users can quickly locate and access specific categories of text based on tags, without spending time browsing the entire dataset. Label classification also helps organize and manage large amounts of text data, making it easier to process and maintain.

By classifying text labels, one can better understand and grasp the structure and content of the text. This helps with the execution of information retrieval and text analysis. Based on specific tags, information related to topics or fields can be summarized and statistically analyzed from text with specific tags. In addition, the results of label classification can also be used for further data mining and machine learning to extract deeper insights and knowledge. Based on the results of label classification, personalized recommendation content can be provided to users. By analyzing user preferences and interests, relevant text content can be recommended based on the tags they are interested in, improving user satisfaction and experience.

Technological convergence framework

In this study, a framework for English translation text classification is constructed, and it is combined with multimedia network technology with transfer learning. This framework uses the BERT model, Att-BILSTM model and TLCM model. First, we can use the BERT model as a feature extractor. The BERT model is a pretrained model based on the transformer architecture that can learn contextual word representations. We can call the pretrained BERT model and encode the input English text into a fixed length vector representation. Then, we can use the Att-BILSTM model as a classifier. The Att-BILSTM model is a model that combines an attention mechanism with a bidirectional long short-term memory (bidirectional LSTM) network, which can capture contextual information and semantic features in text. We use the output of BERT as input of the Att-BILSTM model to train the model for text classification tasks. Finally, we introduce the idea of transfer learning and use the TLCM model to transfer the model. The TLCM model is a transfer learning method. The model trained in the source domain can be transferred to the target domain to reduce the training workload in the target domain. We can use the TLCM model that is pretrained on other tasks for English translation text classification tasks. The overall framework is as follows.

1) Data preprocessing: Before using this framework, we need to preprocess the text data. This step may include text segmentation, stop word removal, stemming and special character processing. These steps can help improve the accuracy and efficiency of the model.

2) Feature extraction: The BERT model outputs vector representations of each word, and we can use these word vectors to construct sentence vectors of the text. A common method is to calculate sentence vectors by averaging all word vectors. Another method is to use a self-attention mechanism to obtain sentence-level representations. Researchers can choose an appropriate method according to their specific needs.

3) Additional attention mechanism layers, such as multihead attention or self-attention layers, can be added to improve the model’s ability to model context and sentence structure information. When using transfer learning, fine-tuning the TLCM model that has been trained on other tasks is consider to adapt it to more target domain data.

4) Model training and parameter adjustment: In the training process, the cross-validation method can be used to select the superparameters of the model, such as the learning rate, batch size and number of cycles. At the same time, validation sets can be used to monitor the performance of the model and perform early stopping.

5) Model evaluation and optimization: After the model training is completed, the test set is used to evaluate the model. Metrics such as accuracy, recall, and F1-score can be calculated to measure the performance of the model. If the performance is not ideal, the model architecture can be adjusted, the hyperparameters can be optimized, or training data can be added. The pseudocode of the whole system design is shown in Table 2 below.

Table 2 Pseudocode of multi label English translation text classification algorithm.

Input: Training parameter set T = {(wk, yjk)Lj = 1}k = 1, model parameter θ.	
1:	W = {wk = 1/N | k = 1, 2, …N}	
2:	repeat	
3:	  for all w = {(Wk, yjk )Nj = 1}k = 1∈D do	
4:	    for l in range(N)	
5:	      wk = {w1, w2, …, wn}	
6:	      Tk = Att-BILSTM(wk)	
7:	      for o in rang (O)	
8:	        Xko = transformer(Tk)	
9:	      end for	
10:	      for p in range (p)	
11:	        ht1 = TLCM(Xko)	
12:	        ht2 = BERT(Xko)	
13:	        ht3 = [ht1,ht2]	
14:	      end for	
15:	      for q in range(Q)	
16:	      vq = Conv(ht1)	
17:	      Fq = Maxpooling(vq)	
18:	      end for	
19:	    M = Concat(Fq , htp)	
20:	    A = Attention(M)	
21:	    W2 = Fully Connected (A, Dropout)	
22:	    ykj = δ (w2)	
23:	  end for	
24:	Calculate the gradient of each parameter	

Experimental design and result analysis

Experimental results

To verify the effectiveness of the model proposed in this article, 10,000 online English translation samples were used in the experiment. These samples were sourced from the Text Classification Competition Dataset of the Modern Informatics Branch of the Chinese Translation Society. Each English translation data sample was screened using the labels from the previous section, and an uncertain number of labels ranging from 0 to 4 was set. This is a typical multilabel classification task dataset and a typical Chinese small sample dataset.

This study mainly focuses on transfer learning in the multimedia network technology environment. We have integrated the content of text label classification and transfer learning from the BERT model, Att-BILSTM model, and TLCM model to establish a new English translation text classification system. In order to verify the differences between this system and other systems, as well as to explore its accuracy and applicability, we will verify these models separately. On the basis of data preprocessing, we conducted experiments from 100 training sessions to 5,000 training sessions on transfer data. Due to the different trends in the calculation results of different models, it is difficult to distinguish if all the results are compared in one graph. Therefore, we conducted comparative analysis separately. The first step is to conduct a comparative analysis between this research system and the BERT model. Please refer to Fig. 6 for details.

Figure 6 Comparison with BERT model.

From Fig. 6, it is not difficult to observe that the classification system proposed in this study has a similar trend to the BERT model at the macro level: as the number of training iterations increases, the accuracy of the English translation continuously decreases, and after exceeding the threshold, there is an upward trend until stabilization occurs. When the number of training data is 100, the classification results of the BERT model are more accurate (approximately 2% higher) than that of the classification method proposed in this study. This is because the classification method designed in this study is based on the combination of multiple models and improvements. When there is less training data, there is less flow layer data, making it difficult to leverage the advantages of this research method. However, as the amount of training data continues to increase, the method designed in this study will remain in an improved state, with a maximum improvement of 28%. Next, a comparative analysis is conducted between the proposed method and the Att-BILSTM model, as shown in Fig. 7.

Figure 7 Comparison with the Att-BILSTM model.

From Fig. 7, we can observe that there is a fundamental difference in accuracy change trends between the Att-BILSTM model and the BERT model in terms of classification. The Att-BILSTM model increases from a minimum of 30% to 71% and then stabilizes in the range of [66, 73]. However, the maximum value never exceeds 86% in this study. Next, a comparative analysis is conducted between the proposed method and the TLCM model, as shown in Fig. 8.

Figure 8 Comparison with the TLCM model.

From Fig. 8, we can clearly observe that although the calculation results of the TLCM model continue to improve in the early stage, after the amount of data becomes too large, the performance declines before it stabilizes. This may be because the number of English translation words has increased, and more obscure words and sentences are included. Thus, the TLCM model cannot correctly complete transfer learning. The accuracy of the TLCM model in the range of [600, 1,800] and [2,100, 2,500] exceeds that of the method proposed in this study, which needs to be further studied in the future.

Impact analysis

In the previous section, it was verified that the classification method proposed in this study has a high accuracy. Next, we analyze the influencing factors that affect the accuracy of this method to explore its applicability in practical situations. It is known that the parameters that can affect the accuracy of the classification method are the learning rate, batch size and number of training rounds. The impact of these three factors on the accuracy is shown in Fig. 9.

Figure 9 Impact analysis.

(A) Comparison of BERT models; (B) Att-BILSTM model comparison; (C) TLCM model comparison.

(a) Comparison of BERT models (b) Att-BILSTM model comparison.

(c) TLCM model comparison.

According to the results in Fig. 9, we can observe that learning rate, batch size, and number of training rounds have different effects on the classification results. The number of training rounds has the most significant impact on the results, reaching 46% of the highest accuracy. Next is batch size, which can improve results within a 44% accuracy range. The impact of learning rate on results is relatively small, only bringing an accuracy improvement of 18%. When the learning rate requirement is low (less than 15%), the improvement effect on the results gradually weakens as the learning rate increases. However, when the learning rate exceeds this threshold, its impact on the results gradually increases until it reaches its maximum value. The impact of batch size on the results shows a continuous increasing trend. As the amount of data gradually increases, this classification method can continuously learn from the data and fluctuate within the range of [2,500,600]. The increase in training rounds shows a linear relationship with the improvement in results in the early stages.

However, after more than five rounds, increasing the number of rounds resulted in a smaller improvement in the results and tended to stabilize. This indicates that the number of training rounds is directly related to the classification results, and at least five rounds of training are required to achieve a relatively stable state. Although there has been some progress compared to other models, further research and adjustments are still needed. In summary, based on the analysis in Fig. 9, we can conclude that learning rate, batch size, and number of training rounds have different degrees of influence on classification results. In order to achieve better classification results, we need to choose appropriate learning rates, batch sizes, and training rounds based on specific situations, and make appropriate adjustments. In addition, it is necessary to pay attention to the moderate selection of training rounds to avoid overfitting and a decrease in generalization performance.

Conclusion

This study achieved a series of encouraging results by applying transfer learning based English translation text classification methods in the multimedia network technology environment. In the field of translation text classification, we have validated the potential of transfer learning in improving classification accuracy and efficiency. The experimental results indicate that traditional machine learning algorithm based classification methods have certain limitations in processing translated texts. In contrast, our proposed transfer learning based method (BATCL) significantly improves the accuracy of classification. By applying pre-trained models to translation text classification tasks, we achieved better performance. The BATCL model performs well in handling various types of translated texts. Whether it’s news reports, academic papers, or social media comments, our method can accurately distinguish the category of text. The BATCL model has shown advantages in both accuracy and efficiency. Comparing our method with traditional machine learning algorithm based methods, we found that our method can achieve better results in translation text classification tasks. This study successfully improved the accuracy and efficiency of English translation text classification in the multimedia network technology environment by applying transfer learning based methods. Our research findings validate the potential of transfer learning in processing translated texts and provide guidance for further improvement and optimization of related algorithms. In the future, we will continue to delve into the application of transfer learning methods in other fields to promote the development and application of artificial intelligence technology.

Supplemental Information

Supplemental Information 1 Algorithm code.

Supplemental Information 2 Raw data.

Additional Information and Declarations

Competing Interests

Author Contributions

Data Availability

The authors declare that they have no competing interests.

Danyang Zheng conceived and designed the experiments, performed the experiments, analyzed the data, performed the computation work, prepared figures and/or tables, authored or reviewed drafts of the article, and approved the final draft.

The following information was supplied regarding data availability:

The code of multi label English translation text classification algorithm is available in the Supplemental File.

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
