# Peer review of "Transfer learning-based English translation text classification in a multimedia network environment"

_PeerJ Computer Science, doi:10.7717/peerj-cs.1842_

## Round 0.1 · original submission · Major Revisions

I checked all concerns received from three reviewers. One of the reviewers is against this paper for publication due to weak novelty and model framework, while the other two reviewers suggest a revision. In my opinion, comments like "weak novelty" are not appropriate, so it is better to recommend a major revision for further consideration.

Reviewer 1 ·

Basic reporting

The article claims on the work on the topic “Transfer Learning-Based English Translation Text Classification in a Multimedia Network Environment”.
- The research literature can be improved with the recent research studies and many recent and advanced works are already there compared to this research.

Experimental design

- When coming to the novelty, author should still improve. Article is still lacking in this part.
- Author has not given more importance on how the model has developed and the article is lacking in the framework explanation.

Validity of the findings

- In the output figures I can see the author has mentioned “This Research Model”, rather author can mention the name of the proposed model, also when author is mentioning the Max and Min in the same figure, it should be mentioned with %.

- Inference from the output is poor and proper justification is not provided for the output figures. Author needs to substantially improve this section and need to give a proper justification why the graph is coming like that and why there is fluctuation in accuracy graphs etc.

Additional comments

No Comments

Reviewer 2 ·

Basic reporting

There are some major comments that authors should address in the main text:
1- Please extend Section 1 by introducing the main problem statement, novel work, and existing main contributions.
2- Which dataset was applied for English Translation Text Classification? Please explain exactly.
3- The TLCM framework should be illustrated with existing components.

Experimental design

Please design a flowchart for Label classification method.

Validity of the findings

Authors should extend evaluation results for the proposed method and compare them with other state-of-the-art case studies.

Reviewer 3 ·

Basic reporting

In this paper, the overall quality of the paper is good. The authors provided adequate explanations for the incentive, motivatin, background of the topic, and explained their approach to solve the problem of using transfer learning for translation task. The authors also used adequate tables and figures to illustrate their findings intuitively and visually.

Experimental design

The experiment design overall is good. The topic of research is aligned with the scope off this journal. The research question of this paper is well defined, relevant & meaningful. The authors stated how research fills an identified knowledge gap, which is the need for powerful translation models. The authors also performed a rigorous investigation on existing methods to ensure a well-established research.

Validity of the findings

1. For figure 4, it's unclear what each of the step represents. Recommend the authors add a description of what each step is, the change of dimensions, etc. to help the readers understand the process better.
2. in figure 7, there's a significant difference between the proposed model and Attention-based BiLSTM. Can you explain the difference in behavior?
3. Looks like figure 6-8 are showing the same performance curve of the proposed model and different other models for comparison. You can put them all on the same figure to give a more intuitive demonstration. Also, for fig 6, looks like the curve for the proposed model is slightly different in terms of resolution and smoothness.
4. The learning rate vs impact rate should be a bell curve, where there should be an optimal learning rate. Too high or too low should result in sub-optimal results. Can you extend the x-axis to show the whole curve of when learning rate goes to 100?
5. i'd like to see a tabluar presentation of the results. While the figures are straghtforward, they do not provide the actualy numbers and it's important to present them.

---

## Round 0.2 · accepted · Accept

All previous reviewers suggest this revised paper can be accepted.

Reviewer 1 ·

Basic reporting

Authors has included the required changes; hence it can be accepted for possible publication.

Experimental design

Authors has included the required changes; hence it can be accepted for possible publication.

Validity of the findings

Authors has included the required changes; hence it can be accepted for possible publication.

Reviewer 2 ·

Basic reporting

Author added improvements and modified the papers in this revision.

Experimental design

Author added improvements and modified the papers in this revision.

Validity of the findings

Author added improvements and modified the papers in this revision.

Additional comments

Author added improvements and modified the papers in this revision.

Reviewer 3 ·

Basic reporting

The paper is in good condition.

Experimental design

The experimental design is adequate

Validity of the findings

Findings are valid